# Kinetic Isotope Effect in the Unfolding of a Protein Secondary Structure: Calculations for Beta-Sheet Polyglycine Dimers as a Model

**DOI:** 10.3390/biom15010092

**Published:** 2025-01-09

**Authors:** Alexey O. Yanshin, Vitaly G. Kiselev, Alexey V. Baklanov

**Affiliations:** 1Institute of Chemical Kinetics and Combustion SB RAS, 3 Institutskaya Street, Novosibirsk 630090, Russia; alexei.yanshin@mail.ru (A.O.Y.);; 2Department of Physics, Novosibirsk State University, 1 Pirogova Street, Novosibirsk 630090, Russia

**Keywords:** protein, unfolding, hydrogen bond, kinetic isotope effect, polyglycine dimer, completely loose transition state

## Abstract

In the present work, we performed calculations of the kinetic isotope effect (KIE) on H/D, ^14^N/^15^N, ^16^O/^18^O, and ^12^C/^13^C isotopic substitution in the dissociation of beta-sheet polyglycine dimers of different lengths into two monomer chains. This dissociation reaction, proceeding via breaking of the interchain hydrogen bonds (H-bonds), is considered to be a model of unfolding of the secondary structure of proteins. The calculated strengthening of the interchain hydrogen bonds N−H⋯O=C
due to heavy isotope substitution decreases in the row H/D >> ^14^N/^15^N > ^16^O/^18^O > ^12^C/^13^C. The KIE for H/D substitution, defined as the ratio of the rate constants $\frac{k(H)}{k(D)}$, was calculated with the use of a “completely loose” transition state model. The results of the calculations show that a very high H/D isotope effect can be achieved for proteins even with moderately long chains connected by dozens of interchain H-bonds. The results obtained also indicate that the heavy isotope substitution in the internal (interchain) and external H-bonds, located on the periphery of a dimer, can provide comparable effects on secondary structure stabilization.

## 1. Introduction

Proteins in their native state are key agents of life, having a unique functionality in the cells of all known organisms. The unfolding of a native structure, defined as denaturation, results in the loss of a protein’s functionality, which can cause the death of cells and organisms. Thermal heating is the most widely used working factor of denaturation. Therefore, agents providing thermostabilization to proteins are of great interest for fundamental studies and applications. More specifically, the use of heavy water (D_2_O) as an agent for slowing down biological processes has been known for a long time [1]. A large amount of related data were reviewed recently in a monograph by Chen [2]. Guild and van Tubergen were probably the first to reveal the thermostabilization of the native state of protein in D_2_O in comparison with that in H_2_O [3]. These authors studied the kinetics of the heat inactivation of enzyme catalase in both solvents and found the protein to be more stable in D_2_O. This thermal stabilization was attributed to the kinetic isotope effect (KIE) in the heat denaturation process. The authors [3] found that the deuteration responsible for KIE proceeds very fast. On the basis of this fact, they attributed the observed heat denaturation to the dissociation of weak interchain hydrogen bonds. This result is in line with the important role of hydrogen bonds in determining the structures of folded states of proteins [4,5]. Then, the thermostabilization of proteins with different functionalities by heavy water has been frequently reported [6,7,8,9,10,11,12,13]. Similar stabilization has also been observed for the β-turn structure in polypeptides [14]. Note that the effect upon the change from light water (H_2_O) to D_2_O is important for health-related issues. Inter alia, this effect results in an increase in the thermostability of various vaccines, including the oral polio virus vaccine [15], the influenza virus A and B vaccines [16], the Newcastle disease virus vaccine [17], and the measles vaccine [18]. All of these issues were reviewed in the recent monograph by Chen [2]. One of the considered mechanisms of this D_2_O effect is thermal stabilization due to the KIE in protein denaturation in the viral capsid of a vaccine [2]. The stabilization of the protein structure in D_2_O can result in a very essential change in biological functionality. An instructive example is the thermal preferendum of *Drosophila melanogaster* flies [19]. When fed with sugar water enriched with D_2_O, these flies demonstrated not only heat resistance but also thermophilicity [19]. This effect was also assigned by the authors to a KIE in protein denaturation [19]. Moreover, the biological system discriminates not only the isotopes in a hydrogen/deuterium pair but other common elements as well. This discrimination manifests itself as an enrichment or depletion of heavy isotopes in different organisms, examples of which are reviewed by Li and Snyder [20]. With the aid of high-resolution mass spectrometry, these authors also found that the content of heavy isotopes in the cellular metabolites of yeast cells declined during chronological aging [21]. This decline was accompanied by a decrease in cell viability. These facts were attributed to heavy isotope accumulation in the heavy isotope sink (HIS) metabolites due to the KIE. Note that the proteins were proposed as candidates for such HIS metabolites [21]. Li and Snyder also found that deuterium enrichment through heavy water uptake extends the chronological lifespan of yeasts by up to 85% with a minimal effect on growth [21]. On the basis of their data, Li and Snyder hypothesized that the higher abundance of deuterium and other heavy isotopes promotes the longevity of different organisms, including humans [20]. The distinct effects of ^15^N-enrichment on the proteomes of organisms were detected by Filiou et al. [22]. These authors investigated the ^15^N-isotope effect on *Escherichia coli* cultures grown in either unlabeled (^14^N) or ^15^N-labeled media. Andriukonis and Gorokhova revealed profound changes in the growth kinetics of the green alga *Raphidocelis subcapitata* in ^15^N-enriched media [23]. These effects were attributed to the KIE as well [22,23].

In bond dissociation reactions, the KIE leads to an increase in bond energies ($D_{0}^{0}$) upon heavy isotope substitution due to the lowering of the zero-point energy (ZPE) of corresponding vibrations [24]. For example, estimation of the difference in ZPE values, taking into account the known vibrational wavenumbers in methane CH_4_(CH_3_D) [25], results in strengthening of the H_3_C-D bond compared to H_3_C-H bonds equal to $\delta_{H}^{D}(E_{C-H\;bond}$) ≈ 1.8 kcal/mol. However, hydrogen bonds are usually about an order of magnitude weaker than their covalent counterparts. This, in turn, renders the isotopic effect on the bond strength less pronounced. More specifically, ab initio calculations performed by Scheiner and Čuma for water dimers and trimers estimated the effect of H/D substitution on H-bond energies to be 0.1–0.2 kcal/mol [26].

Therefore, the importance of the KIE in the heat denaturation of proteins motivated us to perform quantitative estimations of these effects for hydrogen bonds in proteins. Recently [27], we have suggested the dissociation of polyglycine dimers of various lengths as a model process for protein unfolding. The schematic structure of these dimers is shown in Figure 1. Each of the two polypeptide chains contains the glycine fragments -NH-CH_2_-C(O)- with the terminal groups CH_3_CO- and CH_3_NH-. The optimized dimer geometry was found [27] to be similar to the antiparallel chain-rippled sheet structure of crystalline polyglycine I described earlier by Moore and Crimm [28]. The dimer consists of *N* similar structural units with a total number of 2*N* interchain hydrogen bonds. The process of dissociation (1) results in the breaking of these interchain bonds, yielding two monomer chains in a planar all-trans conformation.(1)$$Dimer\;\overset{k}{\to }\;2\;Monomer$$

Recently [27], we calculated the rate constant of process (1) using the “completely loose” transition state model of the transition state theory. The obtained results explained the compensation effect in the kinetics of the protein unfolding as well as the nature of the “exotic” very high pre-exponential factors measured experimentally for the heat denaturation of proteins in a free state and for the thermal inactivation of a huge variety of microorganisms [27]. This explanation of experimental data, obtained usually in a water solvent, with the use of a gas-phase theoretical approach allows us to conclude that the properties of the transition state of the dissociating dimer itself define these exotic values of kinetic parameters. This, in turn, stimulated us to apply the same approach for calculating the KIE values in the present work. According to the computational data of [27], both the activation energy and pre-exponential factor of the unfolding rate constant rise upon the increase in the length of the chain to be unfolded (or, equivalently, the number of interchain H-bonds to be dissociated). The experimental data, reviewed in [27], demonstrate a very wide scatter in the values of Arrhenius parameters. This means the wide spread in the number of interchain H-bonds to be dissociated in the limiting step of protein denaturation. Therefore, the KIE on the different chain lengths to be unfolded is of significant interest. In order to estimate KIE values for the unfolding of proteins, the methodology of [27] was applied in the present contribution. More specifically, the “completely loose” transition state model was used to calculate KIE values for polyglycine dimer dissociation (1) upon H/D isotope substitution in the interchain hydrogen bonds (i.e., the primary isotope effect) as well as in the H-bonds located on the periphery of the polypeptide chains (the secondary isotope effect). Using the same model system, we calculated the isotopic effects (H/D, ^14^N/^15^N, ^16^O/^18^O, and ^12^C/^13^C) on the strength of an interchain hydrogen bond N−H⋯O=C in the dimers of polyglycine of various lengths. The influence of the isotope substitution in all hydrogen bonds on the dissociation energy of the dimers was also considered. This change in dissociation energy provides an enthalpic contribution to the KIE value.

## 2. Methods

### 2.1. Quantum Chemical Calculations

The geometries of all structures corresponding to the stationary points on the potential energy surface (PES) of the species studied were fully optimized in our previous work [27] using density functional theory at the B3LYP/6-31G(d) level [29]. All calculations were performed using the Gaussian 16 suite of programs [30]. Zero-point energies and thermal corrections to thermodynamic potentials were computed at the same DFT level of theory. The substitution of isotopes was made for one particular type of atom involved in the hydrogen bond N−H⋯O=C, with the most abundant isotopes left for other atoms. The effect of H/D substitution was studied in the two variants of substitution: (1)—the change H(D) was made in each of the 2*N* interchain hydrogen bonds of a dimer and (2)—the change H(D) was made in all inter- and intrachain hydrogen bonds of a dimer. The influence on the calculated isotope effect of the dispersion correction was investigated for the case of H/D substitution with the use of the B3LYP-D3 approach [31].

### 2.2. Rate Constant Calculations

The rate constant *k* of the reaction (1) was calculated using Transition State Theory (TST) within the approach of a “completely loose” transition state described in detail in our previous work [27]. In the framework of this approach, the reaction coordinate is the distance $R^{\neq }$ between the centers of mass of monomer chains. The transition state corresponds to the two independently rotating monomer chains located at a particular $R^{\neq }$ value. It was shown [27] that the rate constant of reaction (1) reads as:(2)$$k_{TST}=\frac{\kappa T}{h}\cdot \frac{Q_{pd}\cdot Q_{rot,Mon}^{2}\cdot Q_{vibr,Mon}^{2}}{Q_{rot,Dim}\cdot Q_{vibr,Dim}}\cdot e^{-\frac{∆H_{0}^{0}}{\kappa T}}$$

Here, $Q_{rot,Mon}$ and $Q_{vib,Mon}$ are the rotational and vibrational partition functions of a monomer, and $Q_{rot,Dim}$ and $Q_{vibr,Dim}$ correspond to a dimer. Both partition functions were calculated using the rigid rotor–harmonic oscillator (RRHO) approach. $Q_{pd}=\frac{2\cdot \mu \cdot {{(R}^{\neq })}^{2}\cdot \kappa T}{\hbar^{2}}$ is the rotational partition function of the pseudodiatomic species formed by the two point-masses of monomer units located at a distance $R^{\neq }$. $∆H_{0}^{0}$ is the enthalpy of dissociation reaction (1) at *T* = 0 K. The values of $∆H_{0}^{0}$, rotational constants, and vibrational wavenumbers were calculated using DFT, as described above. The distance between the centers of mass of monomers in the transition state was extended in comparison with that in a reactant (≈5 Å) by an extra 2 Å, yielding $R^{\neq }=7$ Å. The rate constants were calculated at *T* = 298.15 K.

## 3. Results

### 3.1. The Types of Hydrogen Bonds in Polyglycine Dimers and Monomers

The structures of polyglycine dimers and monomers of different lengths have previously been optimized using density functional theory (B3LYP/6-31G(d)) [27]. Figure 2 shows an instructive structural example of the dimer and monomer with the length *N* = 4. Note that the Cartesian coordinates of several dimers studied in the present work are given in the Appendix A. As seen in Figure 2, three types of hydrogen bonds are present in the dimer: an interchain and two types of intrachain H-bonds, viz., internal and external.

The lengths of the bonds of various types differ remarkably, as seen in Figure 3, where the distances of the interchain and intrachain H-bonds are presented for the dimer and monomer with a length of 16 links.

Figure 3 also indicates that all H-bonds, except a few located at the edge, have very similar lengths. The number of intrachain H-bonds in two monomer chains formed in the unfolding reaction (1) is equal to the number of intrachain H-bonds in the dimer. However, Figure 2 shows that the intrachain H-bonds in the dimers are the parts of 7- (C7 H-bond) and 5-membered (C5 H-bond) rings in the dimers and monomers, respectively. This difference results in the shorter interatomic H⋯O distance and shorter intrachain H-bonds in dimers, respectively. This difference in the strength of C5 and C7 H-bonds is well known for peptides and proteins [32].

### 3.2. Effect of Isotope Substitution on the Strength of the Interchain H-Bond and Total Binding Energy of the Secondary Structure of the Dimer

In order to probe the isotope effects on the strength of interchain hydrogen bonds, we calculated the difference between the enthalpies $∆\left(∆H_{0}^{0}\right)$ of reaction (1) upon isotopic substitutions H/D, ^14^N/^15^N, ^16^O/^18^O, or ^12^C/^13^C, when a particular atom of H-bond N−H⋯O=C is replaced with its heavier isotope. Figure 4 shows the computational results obtained for the two fashions of isotopic substitution: Figure 4a—the substitution occurs in each moiety forming an interchain H-bond, and Figure 4b—in those related to all inter- and intrachain H-bonds.

The enthalpy $∆H_{0}^{0}$ of reaction (1) comprises two contributions, viz., the electronic energy and zero-point vibrational energy (ZPE). The former component is naturally independent on the isotopic composition. On the contrary, the ZPE does vary upon the isotopic substitution due to mass-dependence of the vibrational wavenumbers. Therefore, the KIE of the dimer dissociation reaction (1) stems from the ZPE contributions. The slope of the linear dependence in Figure 4 corresponds to the specific change of $∆\left(∆H_{0}^{0}\right)$ per one link. As one link contains two interchain hydrogen bonds, it is instructive to consider the increment ($\delta_{light}^{heavy}(E_{H-bond}$)), corresponding to the change of $∆\left(∆H_{0}^{0}\right)$ upon the isotope substitution in one interchain hydrogen bond. All discussed values are presented in Table 1.

Recall that the ZPE values presented in Table 1 were calculated within harmonic approximation. It is also instructive to discuss the effects of vibrational anharmonicity, which are the most essential for the hydrogen isotopes. Thus, we estimated the influence of anharmonicity on the strengthening of H-bond $\delta_{light}^{heavy}(E_{H-bond}$) for H/D substitution within the approximation of quasidiatomic H-bond with the Morse potential. The computational details are presented in the Appendix A. The anharmonicity results in a reduction in the $\delta_{light}^{heavy}(E_{H-bond}$) value upon H/D substitution by about 10% as compared with the values in Table 1.

The strengthening of the H-bond for H/D substitution in the interchain H-bonds was also calculated with the use of the B3LYP-D3 approach. According to the results presented in Appendix A, the value of δ(ZPE) per one link is equal to 0.12 kcal/mol which differs from the B3LYP value (see Table 1) by 20%.

### 3.3. Effect of Isotope Substitution on the Rate Constant of Unfolding

Subsequently, kinetic isotope effect (KIE) values for H/D substitution were calculated as a ratio of the rate constants $\frac{k(H)}{k(D)}$ of reaction (1). In turn, the rate constants were calculated within Transition State Theory in the framework of the “completely loose” transition state model (more details are given in the Section 2). The resulting KIE values are presented in Figure 5.

The increase in the KIE with the dimer length stems from the growing number of interchain hydrogen bonds undergoing dissociation. For the largest dimer comprising 16 links (i.e., 32 interchain H-bonds), we obtained the KIE value of $\frac{k(H)}{k(D)}$ = 7.0. Note that the latter value is somewhat lower than its counterpart of 17.0 calculated using only the enthalpic contributions $\delta_{H}^{D}(E_{H-bond}$) from Table 1. This difference indicates a partial compensation of the enthalpic factor by the entropic one. Figure 5 also shows that the calculated KIE value in the case of H/D substitution in all H-bonds, including both inter- and intrachain, is equal to $\frac{k(H)}{k(D)}\approx$60.

The KIE value for H/D substitution in the interchain H-bonds was also calculated with the use of the B3LYP-D3 approach. According to the results presented in Appendix A, the value of the KIE for the longest dimer with *n* = 16 is about 20% higher than that obtained with the non-dispersion corrected B3LYP functional.

Apart from this, we also estimated the KIE values of reaction (1) corresponding to the natural abundance of heavy H, N, O, and C isotopes. Heavy isotope natural abundance for these atoms differs by up to 100 times (see Table 1). Nevertheless, the products of natural abundance and δ(ZPE) corresponding to the isotope substitution in all H-bonds vary for different atom types by no more than three times. Thus, the KIE values estimated for the natural abundance on the basis of the calculated enthalpic contributions correspond to a decrease in the unfolding rate constant by only 0.01% per one structural link containing the two pairs of amino acid residues.

## 4. Discussion

Table 1 shows that the H/D substitution results in a maximal strengthening of the interchain H-bond with a subsequent decrease in a row: H/D >> ^14^N/^15^N > ^16^O/^18^O > ^12^C/^13^C. The strengthening of the interchain H-bond $\delta_{H}^{D}(E_{H-bond}$) ≈ 0.05 kcal/mol is much lower than the strengthening of a covalent C-H bond (recall that $\delta_{H}^{D}(E_{C-H\;bond}$) ≈ 1.8 kcal/mol in the case of methane; see the Introduction). At the same time, H/D substitution in a large number of interchain H-bonds dissociating in the process of protein unfolding can ultimately result in a remarkable enthalpy difference. Our data show that the overall strengthening effect in the case of 32 interchain H-bonds is close to that of one covalent C-H bond. The change in the calculated KIE value (Figure 5, blue graph) with the length of the dimer correlates with the change in the dimer dissociation enthalpy (Figure 4a, black graph). At the same time, the enthalpic contribution to the KIE is partially compensated by the entropy difference.

Apart from this, as seen from Table 1, H/D substitution in all H-bonds of the dimer approximately doubles the heavy-isotope-induced increase in the binding energy of the secondary structure as compared with the case of isotope substitution in only interchain H-bonds. We attribute this extra strengthening to the difference in the strength of breaking intrachain H-bonds in a dimer and newly formed intrachain bonds in monomers (Figure 2). Figure 3 shows that the external intrachain H-bonds in dimers are shorter and, therefore, stronger than those in the monomer chains. Thus, the difference in ZPE values of these two types of intrachain H-bonds also contributes to the increase in the dimer binding energy upon H/D substitution. The same substitution in intrachain H-bonds contributes to the calculated KIE as well. Figure 5 shows that the H/D isotope substitution for all H-bonds of a dimer (H/D substitution in all amide NH groups) leads to a substantially larger KIE ($\frac{k(H)}{k(D)}\approx \;$60) than that occurring upon the breaking of only interchain H-bonds ($\frac{k(H)}{k(D)}$ = 7.0). This H/D substitution of all amide protons corresponds solely to the experimental conditions where isotope substitution occurs in the D_2_O solvent instead of H_2_O. The results obtained allow us to conclude that isotope substitution within external intrachain hydrogen bonds, located on the periphery of a dimer, can contribute essentially to the values of the KIE observed experimentally. Our results show that this contribution is comparable with that of the interchain H-bonds.

This means that H/D substitution in peripheral H-bonds can essentially stabilize the secondary structure of proteins. Therefore, we propose another interpretation of the effect of heavy water on the rigidity of native structure for several protein systems [34]. The authors [34] found a significant increase in the rigidity of native structure in D_2_O and found the characteristic time of the structure tightening to be short on a typical time scale for isotope exchange in internal hydrogen bonds. On the basis of these facts, the authors [34] concluded that this tightening is provided not by H/D exchange in the intramolecular H-bonds but by some “solvent effect” of D_2_O [34]. At the same time, the authors emphasized that the possible role of rapid deuteration on the periphery cannot be ruled out completely. Our results show that the tightening of the structure can be attributed to KIE upon H/D substitution in the external intrachain H-bonds. These edge H-bonds are easily accessible, and in our model system, they exhibit even somewhat stronger tightening after H/D exchange than the internal interchain H-bonds.

The comparison of the results of the current work on KIE calculations with the experimental data is not straightforward. For comparison, it is necessary to know the number of interchain H-bonds breaking in the limiting step of the experimentally studied unfolding process. This number is usually not known. But this comparison is useful anyway because it gives the basis for estimating the length of the protein chain responsible for the limiting step of the unfolding process of the experimentally studied protein or polypeptide system. Mendonca et al. [35] studied the influence of H/D substitution on the unfolding kinetics of partially folded (beta-turns) polyglutamic acid and found the KIE values at room temperature to be about 10 (see Figure 4 in Ref. [35]). According to our results (line 1 in Figure 5), the similar values of KIE correspond to the unfolding of about 10 links (or 20 pairs of amino acid residues) in our model system. In a similar way, we can make comparison of our calculated numbers with experimental data for the KIE observed for ^14^N/^15^N substitution. Filiou et al. [22] observed a ~30% difference in *Escherichia coli* culture growth rates in ^14^N- and ^15^N-labeled media. The authors [22] supposed that this isotope effect could be due to alterations in the catalytic activity of enzymes or distortions in their spatial structure. We propose that this effect could be due to the KIE of the breaking of several dozens of H-bonds (according to our data, about 60 H-bonds).

We would also like to comment on the use of B3LYP-D3 methodology for the calculations of H-bond strengthening and the KIE values upon H/D substitution. It was found that the results of B3LYP-D3 calculations were very close to those obtained with the B3LYP approach, i.e., the difference in H-bond strengthening and KIE values lies within 20%. The isotope effects considered are mainly due to the energetics of the H-bonds themselves, which, in turn, are mainly of electrostatic nature with only minor dispersion contributions. This conclusion corresponds to the results of Boese [36], who investigated the effect of dispersion corrections on the accuracy of H-bond energy calculations [36] and references therein. The authors [36] compared the calculated H-bond energies with the reference data for 16 simple complexes containing fewer than eight atoms, e.g., the water dimer, where the contribution of van der Waals interactions is minor. They found that the dispersion correction did not improve the accuracy of the H-bond energy calculations. Moreover, the dispersion corrections render the uncertainties of H-bond energetics even higher [36]. Therefore, our conclusions are based mainly on the B3LYP results. A more pronounced effect of dispersion corrections was found by Boese for complexes with bulkier groups [36], where the contribution of van der Waals interactions is higher. The absence of any essential effect of dispersion corrections on our results indicates only a minor (if any at all) effect of van der Waals interactions on the kinetic isotope effect in proteins unfolding.

## 5. Conclusions

In the present work, we computationally studied the effect of heavy isotope substitution (H/D, ^14^N/^15^N, ^16^O/^18^O, and ^12^C/^13^C) on the dissociation kinetics of polyglycine dimers of different lengths. The latter process was proposed to mimic the unfolding of proteins. The dissociation of dimers occurs upon the scission of interchain hydrogen bonds N−H⋯O=C between the polypeptide chains and the simultaneous formation of the new intrachain H-bonds. The strengthening of the H-bonds upon heavy isotope substitution ($\delta_{light}^{heavy}(E_{H-bond}$)) is due to the change in zero-point vibrational energy (δ(ZPE)). The maximal calculated strengthening of the interchain H-bond occurs in the case of H/D substitution with an increment $\delta_{H}^{D}(E_{H-bond}$) ≈ 0.05 kcal/mol. This value is much less than the strengthening of the covalent C-H bond upon H/D substitution (1.8 kcal/mol for the C-H bond in methane). However, a large number of interchain H-bonds in the long protein chain can result in a significant isotope effect on the unfolding rate constant. The isotope substitution in all hydrogen bonds (interchain and intrachain) N−H⋯O=C in dimers approximately duplicates the increase in the dimer dissociation energy δ(ZPE) as compared with the primary isotope effect caused by only interchain H-bonds. In the case of H/D substitution, the ratio of the rate constants ($\frac{k(H)}{k(D)}$) was also calculated with the use of the “completely loose” transition state model. For the largest polyglycine dimer considered, which contains 32 interchain H-bonds, the calculated KIE value (that is, a primary isotope effect) is equal to $\frac{k(H)}{k(D)}\approx \;$7. The KIE value calculated in the case of H/D substitution in all H-bonds, including interchain and intrachain ones, is equal to $\frac{k(H)}{k(D)}\approx \;$60. This means that the secondary isotope effect, provided by H/D substitution in H-bonds other than interchain ones, is comparable with the primary isotope effect provided by H/D substitution in the interchain bonds. This result indicates that the essential stabilization of the protein structure can take place due to isotope substitution in H-bonds located on the periphery of the protein. We expect high KIE values when an essential part of the atoms of a certain type participating in H-bonds are substituted by heavy isotopes. This takes place when D_2_O is used as a solvent or a particular effort is made to achieve a high level of isotopic substitution for N, C, or O atoms. At the same time, profoundly lower effects were found for the natural abundance of the heavy isotopes of H, N, O, and C.

## Figures and Tables

**Figure 1 biomolecules-15-00092-f001:**
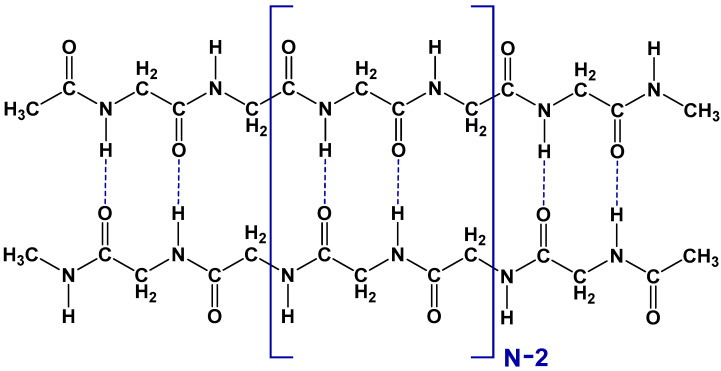
Schematic structure of the polyglycine dimer, consisting of *N* similar structural links. Blue dashed lines indicate interchain hydrogen bonds.

**Figure 2 biomolecules-15-00092-f002:**
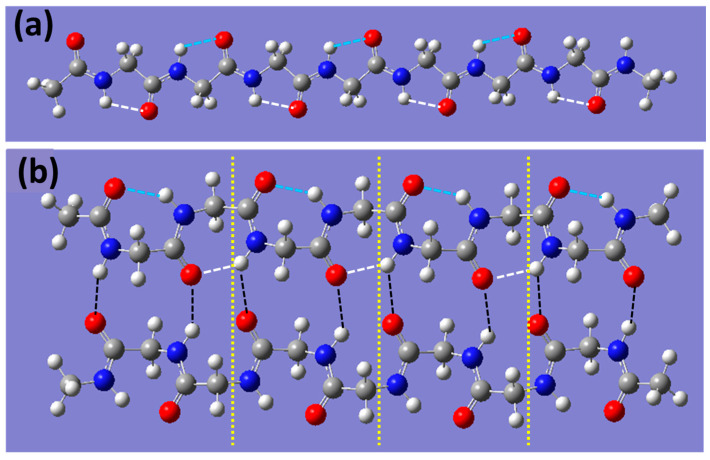
DFT optimized (B3LYP/6-31G(d)) structure of (**a**) polyglycine monomer chain and (**b**) a dimer, consisting of 4 structural units, divided by yellow dotted lines. Black dashed lines indicate interchain hydrogen bonds, and white and blue dashed lines indicate intrachain hydrogen bonds in a dimer (external—blue; internal—white) and monomer. Color code: C—grey; H—white; N—blue; O—red.

**Figure 3 biomolecules-15-00092-f003:**
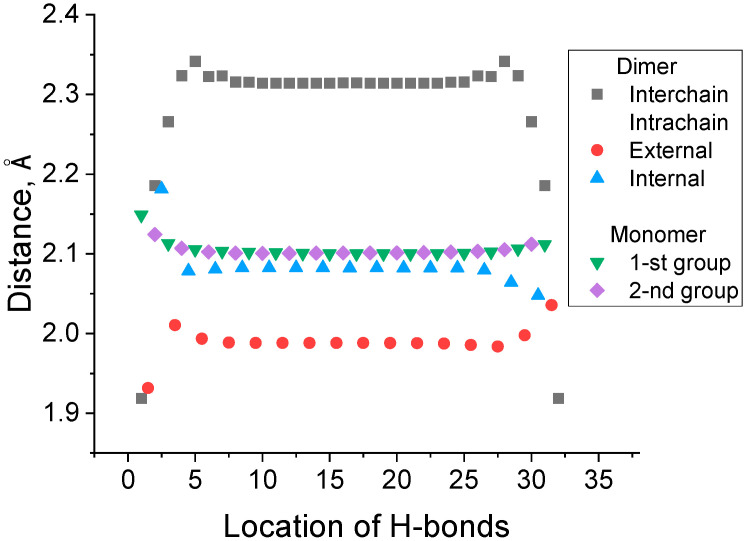
The lengths of the interchain and intrachain H-bonds (H⋯O) in the polyglycine dimer and monomer comprised of 16 links (32 interchain H-bonds). All relevant H-bonds are shown in Figure 2. The location of the interchain H-bonds is denoted by their consequent number from left to right. The location of all intrachain H-bonds is given w.r.t the corresponding number for the closest interchain H-bonds.

**Figure 4 biomolecules-15-00092-f004:**
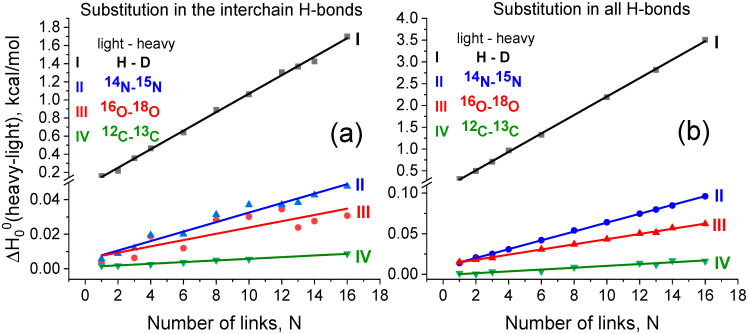
Calculated change in reaction (1) enthalpy $∆H_{0}^{0}$ upon isotopic substitution (light/heavy): (**a**) only in the interchain hydrogen bonds and (**b**) in all hydrogen bonds of the dimers of different length.

**Figure 5 biomolecules-15-00092-f005:**
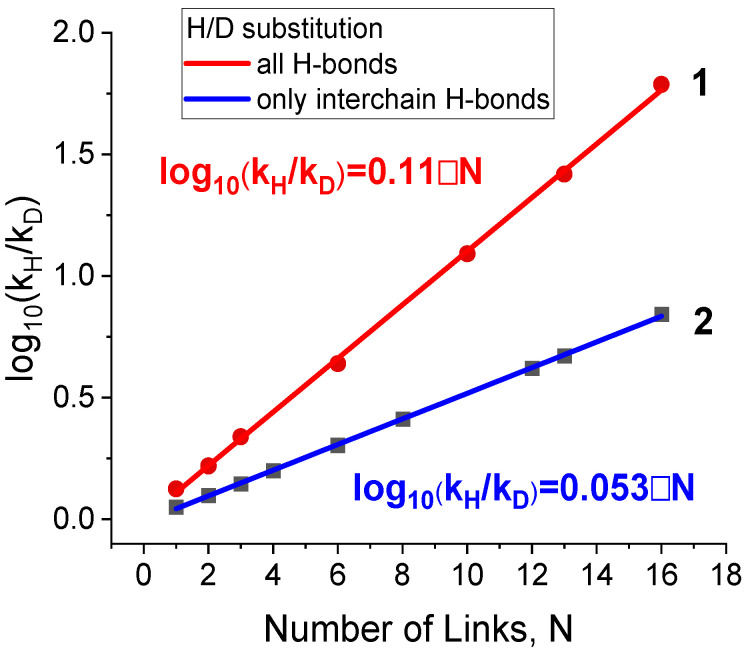
Kinetic isotope effect (KIE), provided by H/D isotope substitution in (1)—all hydrogen bonds (red line) and (2)—the interchain H-bonds (blue line) in the dimers of polyglycine of different lengths. The straight lines were obtained through least squares fitting. The intercepts for both straight lines are equal to 0 within the approximation uncertainty.

**Table 1 biomolecules-15-00092-t001:** Calculated change in vibrational zero-point energy $\delta \left(ZPE\right)$) and strengthening of H-bond
$\delta_{light}^{heavy}(E_{H-bond}$) provided by the light-heavy isotope substitution for atoms, participating in the interchain or in all H-bonds of polyglycine dimers.

Isotope-Substituted Hydrogen Bond	Interchain Hydrogen Bonds		All Hydrogen Bonds	Heavy Isotope Natural Abundance, % [33]
	$\delta \left(ZPE\right)\;per\;link,$ cal/mol	$\delta_{light}^{heavy}(E_{H-bon}),$ cal/mol	$\delta \left(ZPE\right)\;per\;link,$ cal/mol	
N−D⋯O=C	102.3 ± 1.6	51.1 ± 0.8	211.9 ± 1.9	0.0115 (D) (in water)
N15−H⋯O=C	2.74 ± 0.17	1.4 ± 0.1	5.44 ± 0.03	0.368 (^15^*N*)
N−H⋯O18=C	1.8 ± 0.4	0.9 ± 0.2	3.18 ± 0.05	0.205 (^18^*O*)
N−H⋯O=C13	0.48 ± 0.02	0.24 ± 0.01	1.12 ± 0.10	1.07 (^13^*C*)

## Data Availability

Data are contained within the article.

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
