# Peer review of "Kinetic Isotope Effect in the Unfolding of a Protein Secondary Structure: Calculations for Beta-Sheet Polyglycine Dimers as a Model"

_biomolecules, 2025, doi:10.3390/biom15010092_

Round 1

Reviewer 1 Report (Previous Reviewer 1)

Comments and Suggestions for Authors

Authors revised the manuscript according to previous reports.

They carried out new calculations including empirical corrections to dispersive forces (B3LYP-D3) on all studied systems and as expected they observed a modest difference (20%) on final results. In this respect, the most important conclusion is reasonable: "The isotope effects considered are governed by the energetics of H-bonds themselves which are mainly of electrostatic nature with only minor contribution of dispersive interaction."

 The Cartesian coordinates are correctly included in the supplementary materials.

 Nevertheless, the main problem raised by this Reviewer has not been faced.

I agree that "the value of rate constant for dimer dissociation are governed by a maximum of free energy on the reaction coordinate." The authors also state that "the breaking of lower number of bonds doesn’t allow the system to overcome the free energy barrier on the way to the products of dissociation" but this is strictly valid for gas phase processes as that presently reported. In aqueous solution, where experimental quoted data were obtained, when a hydrogen bond breaks two water molecules saturate the dangling HBs. The newly formed water-peptide and the solvent reorganization reduce significantly the transition state and dissociation energies. The same occurs for successive HBs breaking, therefore kinetic and thermodynamic parameters of the overall process are different from the gas-phase dynamics.

Transition state enthalpy grows linearly along the series of dimers as the size increases and for Dim16 it reaches the values around 140 kcal/mol. For Dim32 it is expected to be around 260 kcal/mol and so on. These numbers are too large for these simple peptides in aqueous solution. Peptides like Mon1, Mon2 . . . Mon16 are considered among "intrinsically disordered protein" that depending on the size and experimental conditions associate to form oligomers. Unfortunately, authors do not indicate experimental data on close peptides that could support their data and the matter on transition state energy is a mere speculation.

Anyway, kinetic isotope effect that come from a difference between molecular transition states of two isotopomers, probably, are qualitatively correct thanks to the large compensation error.

Overall, some reliable results are presented and thus the manuscript may deserve publication.

Author Response

Reviewer 2 Report (New Reviewer)

Comments and Suggestions for Authors

The authors used polyglycine dimers of various lengths as a model to investigate the kinetic isotope effect (KIE) in the context of protein folding unfolding events. They performed density functional theory quantum chemical calculations of the strengths of the interchain hydrogen bonds (H-bonds) with and without the substitution of D, 15N, 18O, and 13C isotopes. This work found that KIE kH/kD is in an exponential relationship with the number of glycine residues/H-bonds. Interestingly, the H/D substitution of intrachain H-bonds also contributes to the KIE.

1.     This manuscript presents purely theoretical calculations of the energies of H-bonds in polyglycine dimers and reports no experimentally measured KIE data. The authors state an experimental KIE value for a partially folded polyglutamic acid is similar to the theoretical value for a polyglycine of 10 links. But that comparison is inapt. (1) Glutamic acid has a negatively charge side chain whereas glycine has no side chain. (2) There is no mention of the length of the polyglutamic acid so the authors’ choice of 10 links for the theoretical model is unsupported. (3) The experiment is presumably performed in an aqueous solvent whereas there is no solvent in the DFT calculation, which is another flaw of this method since numerous H-bonds in the solvent are not taken into account.

2.     The authors should modify the title of this manuscript to specify the type of protein secondary structure being studied here: beta-sheet because alpha-helix, for instance, is not represented in the polyglycine dimer model.

Round 2

Reviewer 2 Report (New Reviewer)

Comments and Suggestions for Authors

The authors have satisfactorily addressed my concerns.

This manuscript is a resubmission of an earlier submission. The following is a list of the peer review reports and author responses from that submission.

Round 1

Reviewer 1 Report

Comments and Suggestions for Authors

The manuscript reports on the isotopic effect on the denaturation of protein structure, which is an important issue of biophysics. There is some similarity with a previous paper published by the authors (ref. 27), that deals with the same molecules (dissociation of polyglycine dimers of various length), the same problem (transition state properties), but in the present study the authors analysed the isotopic substitution on the kinetic constant for dimers dissociation and their implication in biochemistry.

They used optimized geometries and force constants obtained in the previous research (ref. 27) to compute new frequencies and partition functions. These computations take few hours therefore the computation efforts of data presented are minimal. This is not clear in the manuscript instead it seems that electronic structure computations are new (see for example section 2.1. Quantum chemical calculations). If there are differences with previous data (geometries and force constants), this information must be explicitly mentioned, otherwise, the authors should indicate that they use the same set of data previously published.

Both in previous and in present manuscripts no supplementary information are reported. It might be useful to include Cartesian coordinates of some dimers to improve the reproducibility of data.

The rate constant k of the dimer dissociation is calculated using transition state theory within the approach of a “completely loose” transition state where the dissociation occurs in a single step. Actually, this very crude approximation may be reasonable for simpler dimers but fails for long chains. The computed energy for the transition state of Dim16 is 140 kcal/mol! Actually, the dissociation occurs in several steps and in each step, there is the breaking of one or two hydrogen bond in a zipper-like mechanism. Entropy/enthalpy compensation claimed in reference 27 and present results are phenomena that occur always in hydrogen bonding dissociation (see for example: J.D. Dunitz, Win Some, Lose Some: Enthalpy-Entropy Compensation in Weak Intermolecular Interactions, Chem. Biol., 1995, 2, 709–712).

I do not know the impact that the present manuscript could have on the scientific community. However, the matter is intriguing, the manuscript is well written and it could deserve publication after the above-mentioned modifications.

Comments on the Quality of English Language

Manuscript well written with a good quality of English.

Reviewer 2 Report

Comments and Suggestions for Authors

In this manuscript, the authors investigated computationally the effect of heavy isotope substitution (H/D, 14N/15N, 16O/18O and 12C/13C) on the dissociation kinetics of the polyglycine dimers of different lengths. This dissociation reaction, proceeding via breaking of the interchain hydrogen bonds (H-bonds), is considered to be a model of unfolding of the secondary structure of proteins. The rate constant k of the reaction is calculated using Transition State Theory (TST) within the approach of a “completely loose” transition state from calculations at the B3LYP/6-31G(d) level of theory in the rigid rotor-harmonic oscillator (RRHO) approach. In this perspective, the manuscript is interesting and the originality of this study stems in the strategy implemented to mimic the unfolding of proteins: the dissociation of dimers occurs upon the scission of interchain hydrogen bonds 𝑁𝐻⋯𝑂 = 𝐶 between the polypeptide chains and simultaneous formation of the new intrachain H-bonds. The strategy implemented was appropriate and the development has a logical build-up with clear explanations. However, I found that there are weaknesses in the choice of the level of theory chosen as well as in some approximations made in the calculation of the rate constant k without any discussion. The two main key points are the following:

            - If the choice of the functional density theory is appropriate for such calculations, the choice of the functional and the basis set used are not. Indeed, the functional used, B3LYP, does not take account the dispersion forces that are nevertheless known to be an important contribution in the energetics of H-bonds. Furthermore, the basis set chosen is not very extended and the basis set superposition error (BSSE) is neither evaluated nor even discussed.

            - The calculations of the rate constant k of the reaction is performed in the rigid rotor-harmonic oscillator (RRHO) approach. The anharmonic effects on the vibrational frequencies and then on the zero-point energies (ZPE) calculations are neither evaluated nor discussed although it is well known that these effects can be significant especially on the weak vibration modes such those of the interchain H-bonds.

Finally, while the H-bonds play a crucial role in the secondary structures of proteins, there is neither discussion nor comparison with others systems of the C5 and C7 well known H-bonds.

In this respect, this paper is not publishable in the present form and I have to reject it.

Reviewer 3 Report

Comments and Suggestions for Authors

This study provides a quantitative evaluation of the kinetic isotope effect in protein unfolding, suggesting that heavy isotope substitution may have a significant impact on protein stability. The findings deepen our understanding of protein stability and unfolding mechanisms, with potential future applications in protein stabilization techniques and biological research using heavy isotopes. Therefore, I generally support the publication of this paper in Biomolecules, but I believe it would be beneficial to address the following points:

1. The calculations are based on a "completely loose" transition state model and do not consider the effects of hydration. The validity of these assumptions and their resulting limitations should be addressed.

2. This paper does not include experimental verification of the calculations. Comparison with results obtained from previous experiments is also limited. Is this a limitation of the study?

3. There is a lack of detailed explanation regarding the calculation conditions and parameter choices, which may make it difficult for non-expert readers to fully understand certain parts of the paper.

Round 2

Reviewer 1 Report

Comments and Suggestions for Authors

Final Review report

I carefully re-read the manuscript and I made an additional in-depth evaluation also of the "R2 second review report.

The manuscript reports calculations on medium size systems with a medium quality quantum chemical level (B3LYP/6-31G(d)). They are fair computations, but the geometry optimizations and force constant calculations have been reported in their previously publication (ref. 27). Now this issue has been correctly addressed. Also the new M06-2X and CCSD(T) computations on simpler models are easily executable and, as I expected, the results do not differ significantly from the B3LYP computation previously carried out. The presented isotopic substitution data require a few hours of computation. So, from a computational point of view, the work is not relevant, i.e. they do not provide a benchmark data for anything.

Further, I noted that the supplementary materials, provided after revision, still do not include Cartesian coordinates as required. They contain energy and gradients up Dim4, that are totally useless!

Although the topic is interesting, the manuscript does not contain relevant data that would justify publication in this form. The accuracy problem of calculations and the anharmonic effect inclusion highlighted by Reviewer R2 are certainly important, but, as shown by the authors, these parameters have a modest effect (20-30% errors) on final results. Much more critical is the "one-step" model that implies inconsistency with appropriate experimental data (200-300% see comments in my second revision). I would encourage authors to take a step forward with respect to the previous publication (ref. 27) for example, to study the large Dim8 or Dim16 in consecutive dissociation of amino acidic residues.

Based on these consideration the manuscript must address these isseues, before publication.

My second Review report

Authors mostly account for my previously comments and appropriate improvements were consequently adopted. They also introduced supplementary materials, new computational data (M06-2X and CCSD(T) on some models) and attempted to estimate vibrational anharmonicity that adds value to the manuscript.

Nevertheless, the main issue concerning the assumed "one-step dissociation" and the related huge activation energy still remain unaddressed. Authors claim that "the literature data for the activation energy of the thermal denaturation of proteins vary in the wide range between about 25 and 200 kcal/mol and reach even the values about 300 kcal/mol". These experimental data refer to entire cells or tissues (epidermis, tendinae, muscle, aorta, cornea, and so on) where thousands of denaturations occur simultaneously. For instance, for large protein such as the bovine serum albumin (67 kDa), the estimated activation energy for denaturation is only 27.6 kcal/mol that is five times lower than the 140 kcal/mol value found for the  very small protein (2 kDa) presently considered. The activation energy for denaturation (140 kcal/mol) seems to be exaggerated and the proposed one-step model may be reliable for small systems up to Dim4, but not for Dim8 or Dim16.

In my opinion, the above mentioned point should be carefully addressed before the manuscript can be published.

Comments on the Quality of English Language

The manuscript requires minor English improvements..

Reviewer 2 Report

Comments and Suggestions for Authors

The modifications made by the authors were really minimal and their explanations and justifications did not convince me. For example, while I agree that the BSSE effects are less with DFT than with post-HF methods, their basis set is really very small, only a double zeta basis set, and it is well known that the BSSE effects become negligible with DFT rather with triple zeta basis sets. Moreover, in these methods and especially with small basis sets, we often have error compensations and it is not because the value is  close to that of a more sophisticated method that we correctly take into account the different effects. That's why I rejected it first and wanted to reject it again.

Reviewer 3 Report

Comments and Suggestions for Authors

The authors have adequately addressed the issues raised by this reviewer.